

# Reliability and validity of an iPhone® application for the measurement of lumbar spine flexion and extension range of motion

Mohammad Reza Pourahmadi[1], Morteza Taghipour[2], Elham Jannati[1], Mohammad Ali Mohseni-Bandpei[2], Ismail Ebrahimi Takamjani[1] and Fatemeh Rajabzadeh[2]

[1] Department of Physiotherapy, School of Rehabilitation Sciences, Iran University of Medical Sciences, Tehran, Iran
[2] Department of Physiotherapy, University of Social Welfare and Rehabilitation Sciences, Tehran, Iran

## ABSTRACT

**Background**. Measurement of lumbar spine range of motion (ROM) is often considered to be an essential component of lumbar spine physiotherapy and orthopedic assessment. The measurement can be carried out through various instruments such as inclinometers, goniometers, and etc. Recent smartphones have been equipped with accelerometers and magnetometers, which, through specific software applications (apps) can be used for inclinometric functions.

**Purpose**. The main purpose was to investigate the reliability and validity of an iPhone® app (TiltMeter© -advanced level and inclinometer) for measuring standing lumbar spine flexion–extension ROM in asymptomatic subjects.

**Design**. A cross-sectional study was carried out.

**Setting**. This study was conducted in a physiotherapy clinic located at School of Rehabilitation Sciences, Iran University of Medical Science and Health Services, Tehran, Iran.

**Subjects**. A convenience sample of 30 asymptomatic adults (15 males; 15 females; age range = 18–55 years) was recruited between August 2015 and December 2015.

**Methods**. Following a 2–minute warm-up, the subjects were asked to stand in a relaxed position and their skin was marked at the $T_{12}$–$L_1$ and $S_1$–$S_2$ spinal levels. From this position, they were asked to perform maximum lumbar flexion followed by maximum lumbar extension with their knees straight. Two blinded raters each used an inclinometer and the iPhone® app to measure lumbar spine flexion–extension ROM. A third rater read the measured angles. To calculate total lumbar spine flexion–extension ROM, the measurement from $S_1$–$S_2$ was subtracted from $T_{12}$–$L_1$. The second (2 hours later) and third (48 hours later) sessions were carried out in the same manner as the first session. All of the measurements were conducted 3 times and the mean value of 3 repetitions for each measurement was used for analysis. Intraclass correlation coefficient (ICC) models (3, k) and (2, k) were used to determine the intra-rater and inter-rater reliability, respectively. The Pearson correlation coefficients were used to establish concurrent validity of the iPhone® app. Furthermore, minimum detectable change at the 95% confidence level ($MDC_{95}$) was computed as 1.96 × standard error of measurement × $\sqrt{2}$.

Corresponding author
Morteza Taghipour, taghipour-morteza@hotmail.com

Peer**J**

**Results**. Good to excellent intra-rater and inter-rater reliability were demonstrated for both the gravity-based inclinometer with ICC values of $\geq 0.84$ and $\geq 0.77$ and the iPhone® app with ICC values of $\geq 0.85$ and $\geq 0.85$, respectively. The $MDC_{95}$ ranged from 5.82° to 8.18° for the intra-rater analysis and from 7.38° to 8.66° for the inter-rater analysis. The concurrent validity for flexion and extension between the 2 instruments was 0.85 and 0.91, respectively.

**Conclusions**. The iPhone®app possesses good to excellent intra-rater and inter-rater reliability and concurrent validity. It seems that the iPhone® app can be used for the measurement of lumbar spine flexion–extension ROM.

**Level of evidence**. IIb.

## INTRODUCTION

Measurement of lumbar spine range of motion (ROM) is routinely used by clinicians in order to assess patients with low back pain (*Saur et al., 1996*). Therefore, clinicians can determine whether an abnormal limitation exists in lumbar spine mobility and can monitor the effectiveness of rehabilitation process (*Bible et al., 2010*). Measurement of lumbar spine ROM can be carried out through visual observation or by using a number of clinical instruments including: motion analysis, flexible curve, radiograph, inclinometer, goniometer, and measuring tape (*Clarkson, 2005*; *Kolber et al., 2013*; *Salamh & Kolber, 2013*). The method or instrument a clinician utilizes to measure lumbar spine ROM may vary and is dependent on biomedical criteria such as accessibility of the instrument, cost, familiarity, easy administration, safety, insusceptibility or insensitivity to external influences, and educational background (*Chaffin, Andersson & Martin, 2006*; *Kolber et al., 2013*).

An inclinometer or tilt meter is an instrument for measuring angles of slope (or tilt), elevation or depression of an object or body segment with respect to gravity (*Nielsen, Chambers & Farr, 2011*). This instrument has been widely used in orthopedic assessments and many studies have examined the reliability and validity of lumbar spine inclinometer measurements (*Newton & Waddell, 1991*; *Ng et al., 2001*; *Nitschke et al., 1999*; *Otter et al., 2015*; *Saur et al., 1996*; *Waddell et al., 1992*). *Saur et al. (1996)* evaluated the reliability and validity of measuring lumbar spine ROM in 44 patients with chronic low back or leg pain using an inclinometer technique. The findings of their study demonstrated that a gravity-based inclinometer is a highly reliable and valid instrument for measuring standing lumbar spine flexion–extension ROM in chronic low back pain patients ($r \geq 0.75$) (*Saur et al., 1996*). In addition, the inter-rater reliability of the gravity-based inclinometer for lumbar spine flexion ROM was good ($r = 0.88$), compared to extension ($r = 0.42$). *Ng et al. (2001)* investigated the intra-rater reliability of lumbar spine ROM in 12 healthy subjects by an inclinometer with the use of a pelvic restraint device. The procedure showed good intra-rater reliability for lumbar flexion ($r = 0.87$), extension ($r = 0.92$), and lateral flexion ($r \geq 0.94$) ROM (*Ng et al., 2001*). In another study, *Waddell et al. (1992)*

assessed the inter-rater reliability of an electronic inclinometer for the measurement of total thoracolumbo–pelvic flexion and extension, isolated lumbar spine flexion, extension, and lateral flexion in 120 patients with low back pain and 70 asymptomatic subjects. They reported good reliability with intraclass correlation coefficients (ICCs) in the range from 0.87–0.95 (*Waddell et al., 1992*). *Newton & Waddell (1991)* examined the inter-rater reliability of lumbar spine flexion ROM in 20 patients with low back pain using a computerized inclinometer, and also reported a good correlation between the results of inclinometer and X-ray measurements ($r = 0.76$). Good reliability and validity of the inclinometer technique for measuring lumbar spine ROM have also been reported in other studies (*Garmabi et al., 2012*; *Mellin, Kiiski & Weckström, 1991*; *Nitschke et al., 1999*). Considering the results of previous studies, it can be concluded that the inclinometer can be used as a reference standard for measuring lumbar spine ROM.

The number of smartphone users is rapidly increasing worldwide, and it has been estimated that this number will rise to 2.5 billion people by 2017 (*Faurholt-Jepsen et al., 2016*). In addition, the advent of smartphones (such as the iPhone® and those that use the Android™) has brought a wide range of clinical measurement applications (apps) within the reach of most clinicians. The vast majority of smartphones have numerous built-in sensors such as accelerometers, magnetometers, and gyroscopes that make the phone capable of detecting joint positions and measuring joint ROM (*Otter et al., 2015*). A number of smartphone based inclinometry apps are now available (iHandy® level, TiltMeter® -advanced level and inclinometer, etc.) which help clinicians measure joint ROM more accurately, easily, and quickly.

To date, several studies have investigated the reliability and concurrent validity of these apps for measuring lumbar spine posture and ROM. *Salamh & Kolber (2013)* assessed the inter-rater and intra-rater reliability, and concurrent validity of a gravity-based bubble inclinometer and iPhone® app (iHandy© level) for measuring standing lumbar spine lordosis in 30 asymptomatic subjects. Good inter-rater and intra-rater reliability was reported for the inclinometer (ICC = 0.90 and 0.85, respectively) and iPhone® app (ICC = 0.96 and 0.81, respectively). The concurrent validity between the 2 instruments was good with a Pearson correlation coefficient ($r$) of 0.86 (*Salamh & Kolber, 2013*). In another study, *Kolber et al. (2013)* evaluated the inter-rater and intra-rater reliability, and concurrent validity of active thoracolumbo-pelvic flexion, extension, lateral flexion, and isolated lumbar spine flexion using a gravity-based bubble inclinometer and iPhone® app (iHandy© level) in 30 asymptomatic subjects. In general, the results of the study indicate a good intra-rater and inter-rater reliability bubble inclinometry (ICC ≥ 0.81) and the iPhone® (ICC ≥ 0.80). The concurrent validity between bubble inclinometry and the iPhone® app was good with the ICC values of ≥ 0.86. However, within-day intra-rater reliability was not examined (*Kolber et al., 2013*). Therefore, the aim of the present study was to assess between-day (intra-rater) and within-day (intra-rater and inter-rater) reliability, minimum detectable change (MDC), and concurrent validity of a gravity-based inclinometer (Vertex) and iPhone® app (TiltMeter© -advanced level and inclinometer) for measuring standing lumbar spine flexion and extension ROM in asymptomatic subjects. The hypotheses of the study were as follows:
**Table 1  Subjects' baseline demographic characteristics.**

| Asymptomatic subjects | n | Age (years) | Body Mass (kg) | Height (cm) | BMI[a] (kg/m$^2$) |
|---|---|---|---|---|---|
| Male | 15 (50%) | 28.70 ± 6.14 | 72.33 ± 10.96 | 176.19 ± 7.67 | 23.02 ± 3.48 |
| Female | 15 (50%) | 27.06 ± 5.24 | 66.08 ± 9.80 | 165.37 ± 7.70 | 23.61 ± 3.52 |
| Total | 30 (100%) | 27.92 ± 6.31 | 67.36 ± 11.77 | 170.07 ± 9.55 | 23.10 ± 3.68 |

**Notes.**

Values are presented as mean ± SD.

[a]BMI: Body Mass Index.

(a) There is no statistically significant difference between the reliability of the 2 instruments for measuring standing lumbar spine flexion and extension ROM.

(b) The iPhone® app (TiltMeter© -advanced level and inclinometer) is a valid and reliable instrument to measure lumbar spine flexion and extension ROM.

## MATERIALS & METHODS

### Study design and subjects

This cross-sectional observational study was conducted between August 2015 and December 2015 (17 weeks). Approval for the study was obtained from the Ethics Committee at the University of Social Welfare and Rehabilitation Sciences (ethical approval number: 801-2-6189) (Tehran-Iran). The sample size used in the study was determined based on previous similar studies (*Kolber et al., 2013*; *Salamh & Kolber, 2013*). *Salamh & Kolber (2013)* calculated the sample size for their study and reported that 28 subjects would be sufficient to achieve 80% power at an alpha level of 0.05. Therefore, 30 asymptomatic adult subjects (males $n = 15$, females $n = 15$) met inclusion criteria and participated in the current study. All of the subjects were identified and recruited by posters, emails and word of mouth from the University and the surrounding community. The inclusion criteria were: (i) no history of low back pain during the last 6 months, (ii) age $\geq$ 18 years (*Kolber et al., 2013*; *Salamh & Kolber, 2013*); (iii) ability to stand and walk independently without an assistive device (*Kolber et al., 2013*; *Salamh & Kolber, 2013*); (iv) no obvious deformity of the spine, pelvis, and lower extremities; (v) the absence of low back pain or lower extremity pain during data acquisition time (*Kolber et al., 2013*; *Salamh & Kolber, 2013*); (vi) no surgical instruments in the spine, and (vii) ability to provide informed consent. Subjects' baseline demographic characteristics are presented in Table 1. An informed written consent was obtained from the subjects before participation.

### Instruments

A standard gravity-based inclinometer (model A–300; Vertex Co., Taiwan) (Fig. 1) and iPhone® model 5 (iPhone® is a trademark of Apple Inc, Cupertino, California, USA) with TiltMeter© -advanced level and inclinometer app (free version; downloaded from Apple's App Store) were used in this study. The TiltMeter© -advanced level and inclinometer app (Fig. 2) is a professional grade angle measurement instrument which measures the degree of tilt of a surface relative to the horizontal plane. The app uses the iPhone's built-in accelerometer and a digital display to show the measured angle.

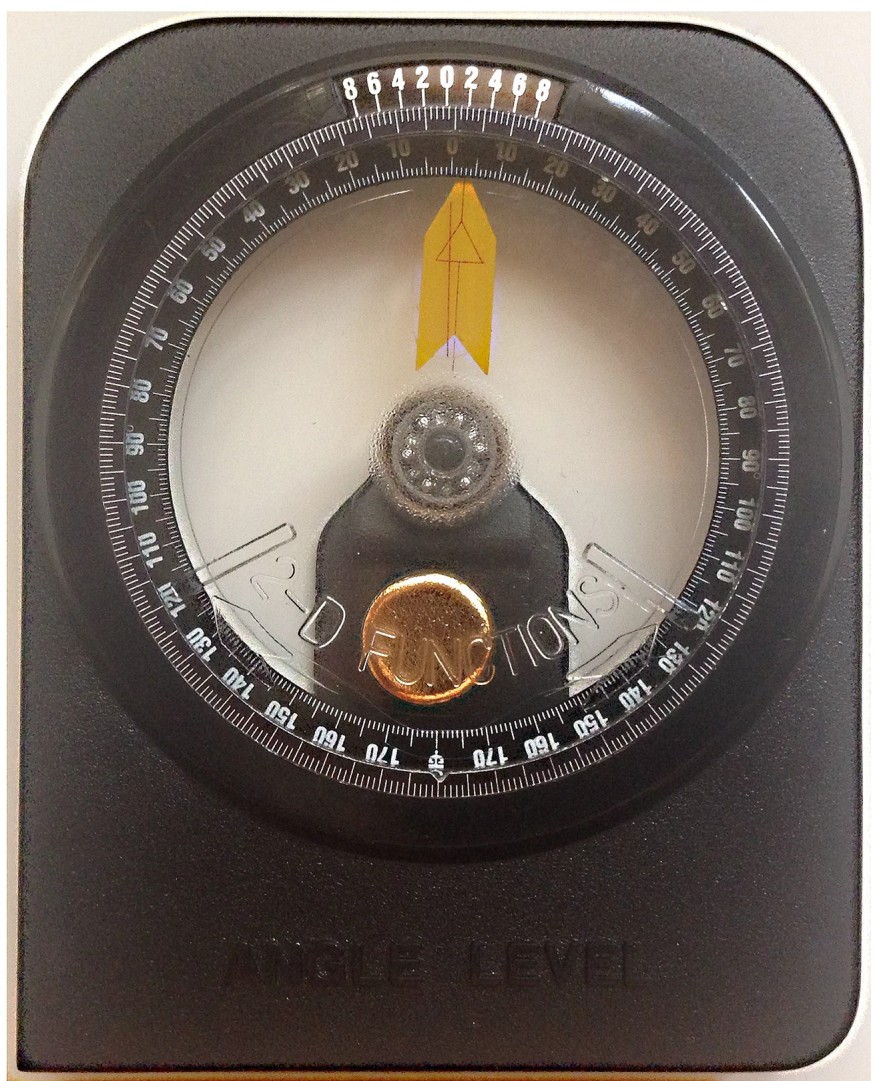

**Figure 1  Standard gravity-based inclinometer (model A-300; Vertex Co).**

## Procedure

Following the recruitment process, the subjects were brought into a physiotherapy clinic at the School of Rehabilitation Sciences (Iran University of Medical Sciences and Health Services) where they performed a warm-up exercise supervised by 3 raters (MRP, EJ, and MT) who were Ph.D. candidates of physiotherapy. The raters had more than 6 years of experience in physiotherapy clinical practice. The warm-up exercise needed approximately 2 min to complete and consisted of lumbo-pelvic rotations in a crook lying position (lower trunk rotation). The physiotherapist asked the subjects to rhythmically rotate the legs about 45° to the right and left. The subjects were also requested to keep their legs together and maintain the soles of the feet on the treatment table while doing the exercise (Fig. 3) (*Kolber et al., 2013*; *Salamh & Kolber, 2013*). In addition, the subjects were familiarized with the exercise through explanation and demonstration before they performed the warm-up. The

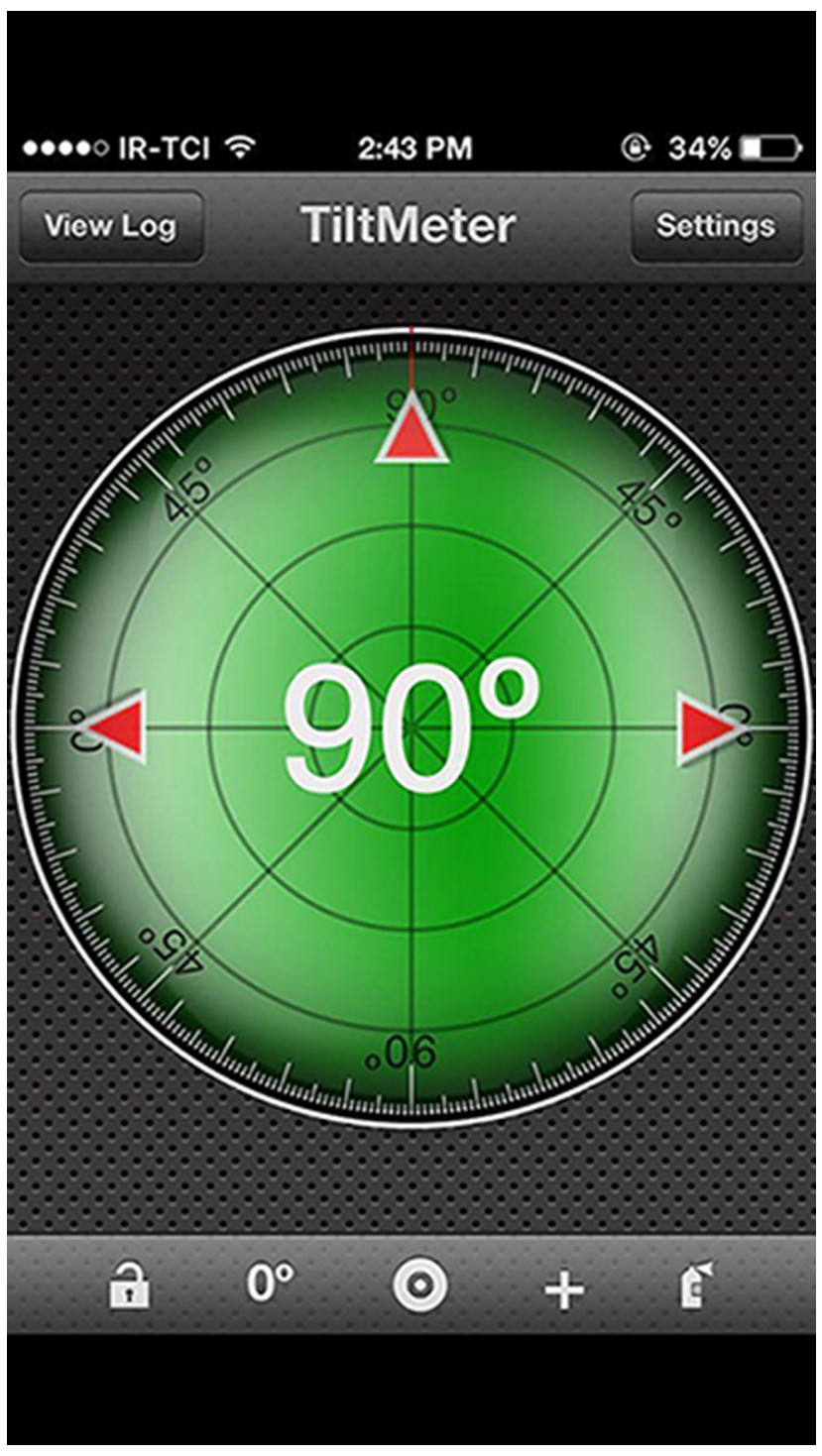

**Figure 2** TiltMeter© -advanced level and inclinometer app.

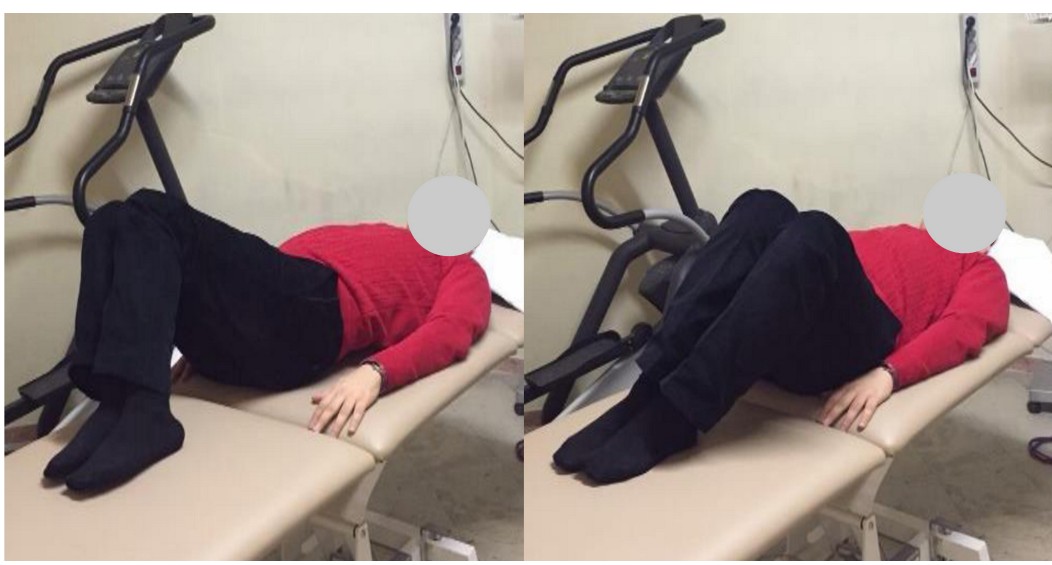

**Figure 3  Lumbo-pelvic rotations in the crook lying position.**

angle of lumbo-pelvic rotation was checked visually by the physiotherapist. In order to have a similar situation for all subjects, they were asked to perform the same warm-up exercise. The goal of the warm-up exercise was to decrease lumbar spine stiffness. However, at present, there is no evidence to support this claim.

Following the warm-up exercise, the subjects were asked to stand in a relaxed position with their feet apart at shoulder width and their arms hanging loosely at their side. The intercrestal (*Kim et al., 2003*) or intercristal line (*Williams & Warwick, 1980*)— the line joining the superior aspect of the iliac crests posteriorly—was used to find the $L_3$ spinous process or $L_3$–$L_4$ spinal level (*Chakraverty, Pynsent & Isaacs, 2007*). Once the $L_3$ spinous process or $L_3$–$L_4$ spinal level was identified, the physiotherapist palpated the spinous processes in the midline and traced them from inferior to superior to find the $T_{12}$ spinous process. The posterior superior iliac spine (PSIS) line was also used to find the $S_2$ spinous process (*Chakraverty, Pynsent & Isaacs, 2007*). Once the $S_2$ spinous process was identified, the physiotherapist palpated the $S_2$ spinous process in the midline and traced it from inferior to superior to find the $S_1$ spinous process. Afterward, subjects' skin was marked at the $T_{12}$–$L_1$ and $S_1$–$S_2$ spinal levels using a black eyeliner (Fig. 4). The inclinometer was placed on the landmarks according to *Waddell et al. (1992)* study (Fig. 5). The iPhone® was placed on the landmarks with contact through the bottom side (Fig. 6). The iPhone® and inclinometer did not require calibration before measurements. All the measurements were obtained by 2 raters (MRP and EJ) in maximum lumbar spine flexion and extension position. Randomization in measurements was not used because the aim of the study was to investigate reproducibility, which requires a consistent physiological status (*Kolber et al., 2013*) In neutral position, the subjects were requested to stand in a comfortable position with their feet apart at shoulder width and their arms hanging loosely at their side. From this position, the subjects were asked to perform maximum lumbar spine flexion followed by maximum lumbar spine extension with their legs straight. Verbal encouragement
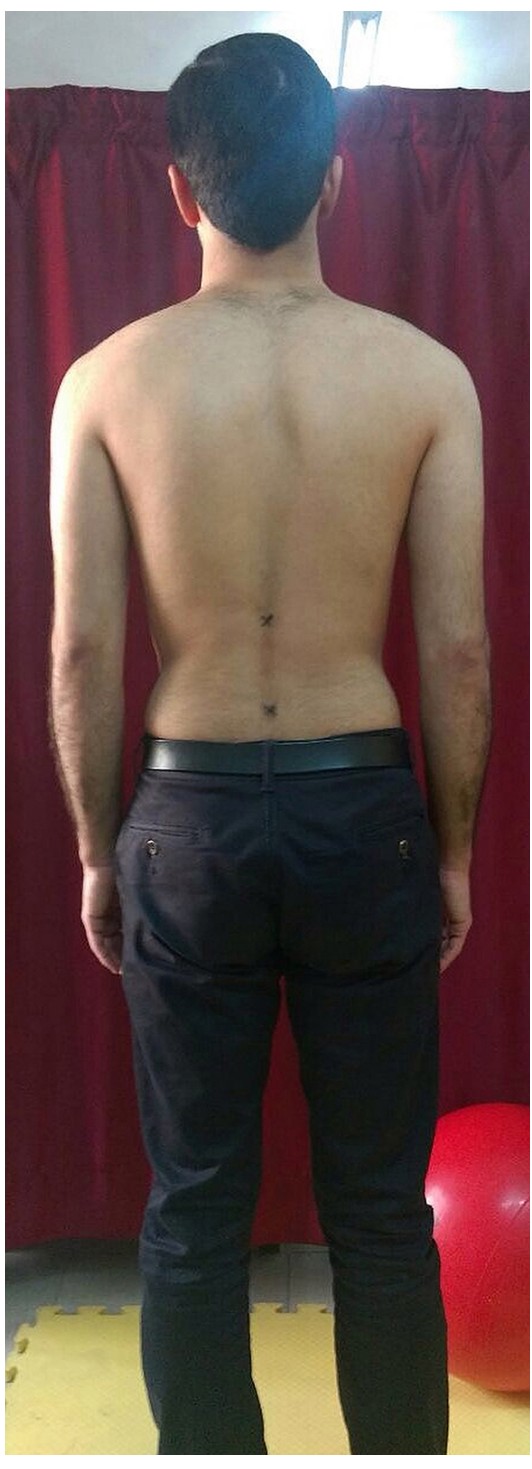

**Figure 4** Starting position for the measurements with landmarks at the $T_{12}$–$L_1$ and $S_1$–$S_2$ spinal levels.

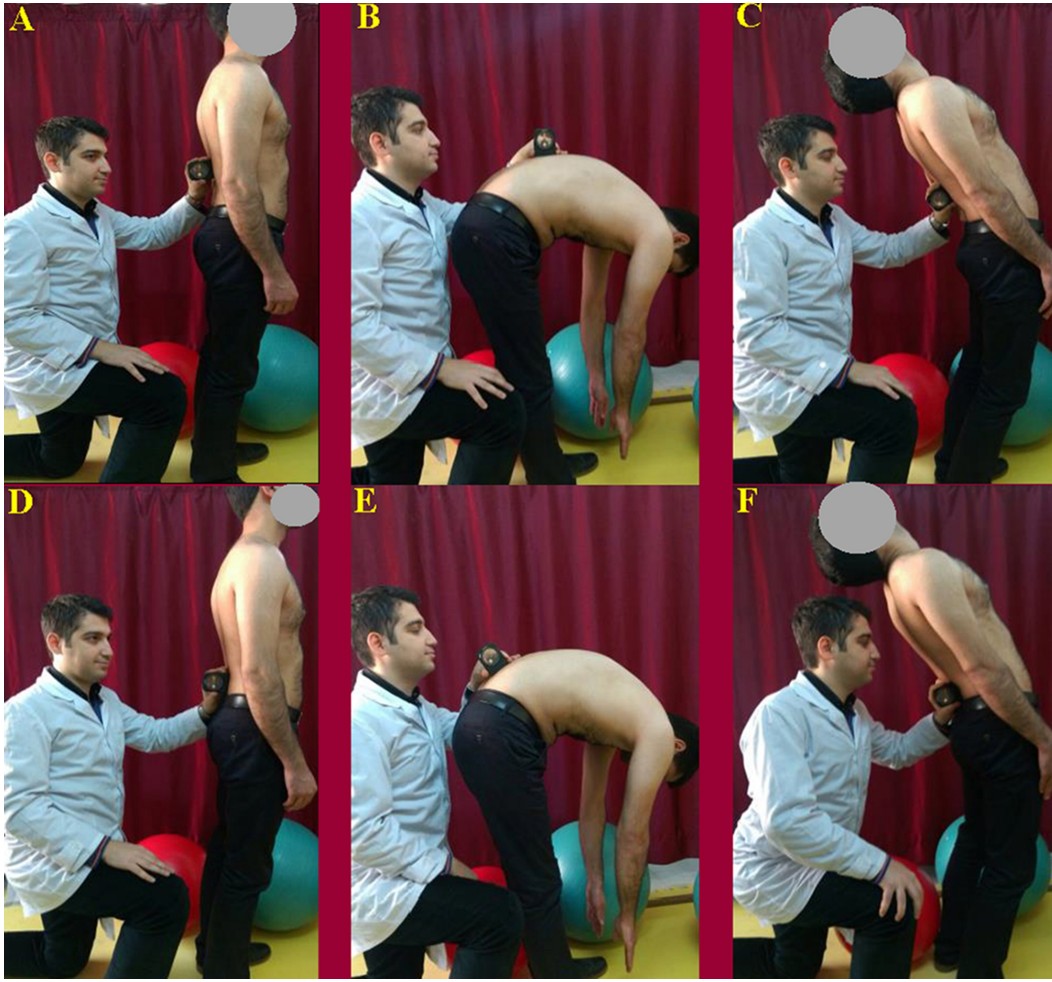

**Figure 5 Measurement of lumbar spine ROM using the gravity-based inclinometer.** (A) Starting position with the inclinometer was placed on the $T_{12}$–$L_1$ spinal level. (B) Maximum flexion was measured at the $T_{12}$–$L_1$ spinal level. (C) Maximum extension was measured at the $T_{12}$–$L_1$ spinal level. (D) Starting position with the inclinometer was placed on the $S_1$–$S_2$ spinal level. (E) Maximum flexion was measured at the $S_1$–$S_2$ spinal level. (F) Maximum extension was measured at the $S_1$–$S_2$ spinal level.

(feedback) was also provided by 2 raters (MRP and EJ) to encourage the subjects to make their maximum effort in order to reach the end of the available range. The inclinometer was first placed on the $T_{12}$–$L_1$ level and then on the $S_1$–$S_2$ level at the extremes of lumbar spine flexion and extension (Figs. 5 and 6). The 2 raters (MRP and EJ) were blinded to the results at the time of measurement, therefore, the movement of the lumbar spine was read directly from the inclinometer by a third person (MT). To calculate total lumbar spine flexion and extension ROM with an inclinometer, the measurement from the $S_1$–$S_2$ was subtracted from the $T_{12}$–$L_1$. This technique of measurement was repeated with the iPhone® using the TiltMeter© -advanced level and inclinometer app. The raters (MRP and EJ) were blinded in this technique as well and the third rater (MT) read the measured angle from the TiltMeter© -advanced level and inclinometer app. After an interval of 2 hours, the procedure was repeated in the same manner.

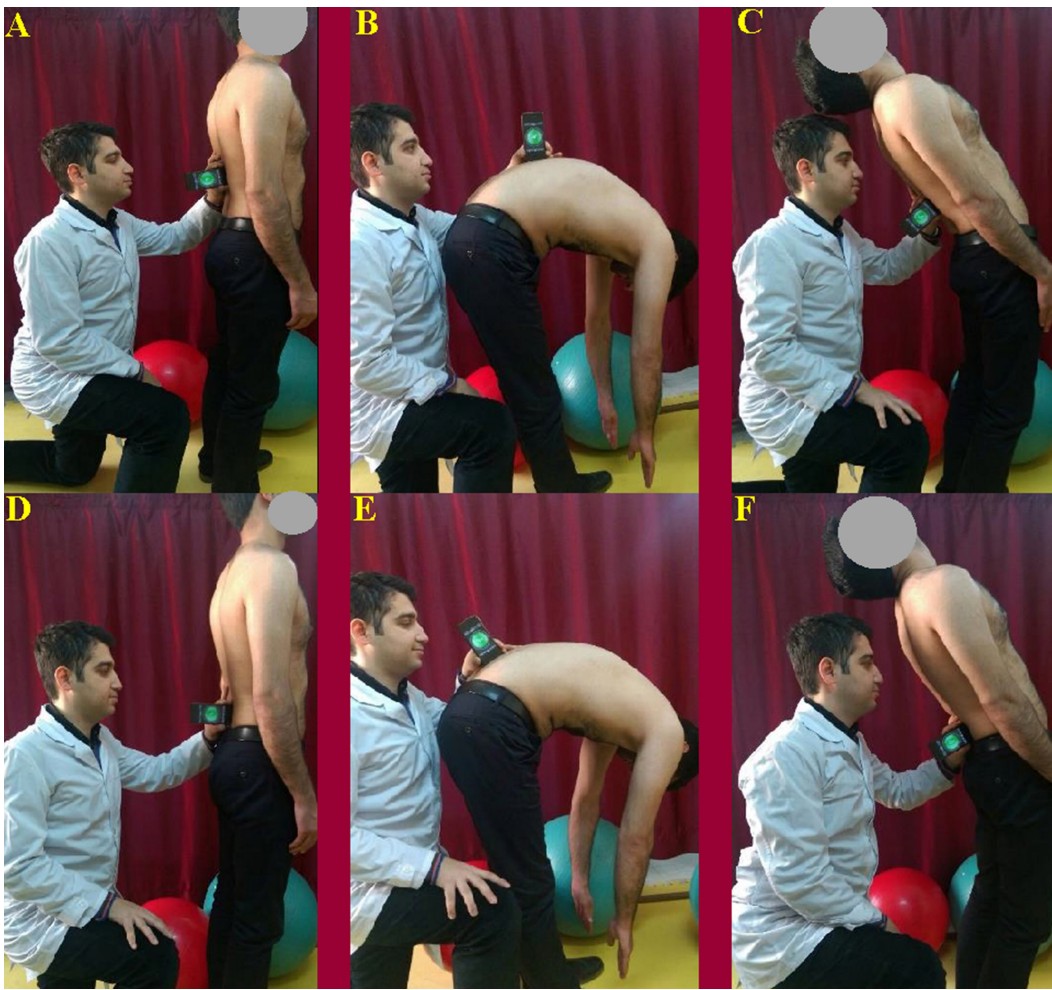

**Figure 6** **Measurement of lumbar spine ROM using the TiltMeter$^{©}$ -advanced level and inclinometer app.** A) Starting position with the iPhone$^{®}$ was placed on the $T_{12}$–$L_1$ spinal level. (B) Maximum flexion was measured at the $T_{12}$–$L_1$ spinal level. (C) Maximum extension was measured at the $T_{12}$–$L_1$ spinal level. (D) Starting position with the iPhone$^{®}$ was placed on the $S_1$–$S_2$ spinal level. (E) Maximum flexion was measured at the $S_1$–$S_2$ spinal level. (F) Maximum extension was measured at the $S_1$–$S_2$ spinal level.

The third session (48 hours later) was started with a 2-minute warm-up exercise (same as the first session), and then the measurements were repeated again. In order to control confounding variables and to improve consistency among the placement of the instruments, one rater (MRP) was responsible for marking the subjects in all 3 sessions.

All the measurements were conducted 3 times by the 2 raters and the mean value of 3 repetitions from each measurement was used for analysis.

## Statistical analysis

All the statistical analyses were performed on a personal laptop using Statistical Package for the Social Sciences (SPSS for Windows release, Version 21.0, Chicago, IL). Prior to the statistical analyses, the Kolmogorov–Smirnov (K–S) test was performed to evaluate the normality of data distribution. For all tests, the statistical significance level was set

**Table 2 Descriptive statistics for gravity-based inclinometer and iPhone® app values.**

| Instrument | Rater | Mean flexion ROM[a] ± SD[b] (Minimum value–Maximum value) | Mean extension ROM[a] ± SD[b] (Minimum value–Maximum value) |
|---|---|---|---|
| Gravity-based inclinometer | A | 52.30° ± 5.82° (40.51°–68.32°) | 18.43° ± 6.26° (5.20°–35.17°) |
| | B | 51.90° ± 4.07° (44.69°–60.21°) | 18.06° ± 6.16° (7.74°–33.18°) |
| iPhone® app | A | 52.00° ± 6.21° (43.94°–70.12°) | 16.50° ± 5.97° (5.21°–34.14°) |
| | B | 51.00° ± 6.18° (41.32°–66.11°) | 16.83° ± 5.87° (4.89°–35.07°) |

**Notes.**
[a]ROM, Range of motion.
[b]SD, Standard deviation.

at $P < 0.05$. Descriptive data are expressed as mean measurement angles ± standard deviations (SD) calculated for each series of measurements (Table 2). The reliability of all measurements was tested using the ICC model (3, k) for the intra-rater analysis and the ICC model (2, k) for the inter-rater analysis. Model (2, k) was also used to assess whether the instrument of choice (gravity-based inclinometer, iPhone® app) can be utilized with reliability and confidence among equally experienced clinicians (*Kolber et al., 2013*; *Salamh & Kolber, 2013*). Using SPSS software, the ICC model (2, k) was computed by selecting the options 2-way random, average measure, and absolute agreement, and the ICC model (3, k) by selecting 2-way mixed and average measure.

The scale from *Bland & Altman (1999)* was used in the classification of the reliability values ($\leq$0.20 poor, 0.21–0.40 fair, 0.41–0.60 moderate, 0.61–0.80 good, and 0.81–1.00 excellent). Standard error of measurement (SEM) was employed to examine the precision of the instruments and was calculated as follows: $SD \times \sqrt{1 - ICC}$. One SEM and 2 SEMs were calculated from the ICC models (2, k) and (3, k) used in this study. One SEM shows that the clinician may be 68% certain that the true measurement value lies within ±1 SEM of measurement from the clinical measurement, whereas, 2 SEMs provide the clinician with 95% of confidence (*McKenna, Cunningham & Straker, 2004*). Furthermore, minimum detectable change at the 95% confidence level ($MDC_{95}$) was computed as $1.96 \times SEM \times \sqrt{2}$ which represents the magnitude of change necessary to provide confidence that a change is not be the result of random variation or measurement error (*Haley & Fragala-Pinkham, 2006*).

The concurrent validity between the gravity-based inclinometer and iPhone® app was evaluated by using the Pearson correlation coefficient ($r$). The correlation coefficient can vary from $-1$ to $+1$. A value of 0 indicates that there is no linear relationship between the two continuous variables. A value greater than 0 indicates a positive relationship; that is, as one variable increases, the other variable also increases (direct relationship). A value less than 0 indicates a negative relationship; that is, as one variable increases, the other variable decreases (inverse relationship) (*Ling et al., 2009*). To calculate the concurrent validity, the rater A (MRP) was the only rater of interest.

The 95% limits of agreement were also calculated as mean difference ± 1.96 × SD (*Bland & Altman, 1999*). After the statistical analysis of data, the results were rounded to 2 decimal places.

**Table 3  Intra-rater reliability of gravity-based inclinometer and iPhone® app.**

| Rater | Measurement | Intra-rater reliability | Gravity-based inclinometer | | iPhone® app | |
|---|---|---|---|---|---|---|
| | | | Within-day (2 h) | Between-day (48 h) | Within-day (2 h) | Between-day (48 h) |
| A | Flexion ROM[a] | ICC[b] (95% CI[c]) | 0.89 (0.77–0.94) | 0.87 (0.74–0.94) | 0.92 (0.84–0.96) | 0.87 (0.73–0.94) |
| | | SEM[d] | 2.60° | 2.65° | 2.41° | 2.99° |
| | | 2 SEMs[e] | 5.20° | 5.30° | 4.82° | 5.98° |
| | | MDC$_{95}$[f] | 7.20° | 7.34° | 6.69° | 7.96° |
| | Extension ROM[a] | ICC[b] (95% CI[c]) | 0.92 (0.83–0.96) | 0.91 (0.82–0.96) | 0.92 (0.83–0.96) | 0.90 (0.78–0.95) |
| | | SEM[d] | 2.30° | 2.30° | 2.25° | 2.55° |
| | | 2 SEMs[e] | 4.60° | 4.60° | 4.50° | 5.10° |
| | | MDC$_{95}$[f] | 6.39° | 6.39° | 6.24° | 7.07° |
| B | Flexion ROM[a] | ICC[b] (95% CI[c]) | 0.85 (0.69–0.93) | 0.84 (0.67–0.93) | 0.92 (0.83–0.96) | 0.90 (0.80–0.95) |
| | | SEM[d] | 2.12° | 2.21° | 2.10° | 2.46° |
| | | 2 SEMs[e] | 4.24° | 4.42° | 4.20° | 4.92° |
| | | MDC$_{95}$[f] | 5.88° | 6.13° | 5.82° | 6.82° |
| | Extension ROM[a] | ICC[b] (95% CI[c]) | 0.88 (0.75–0.94) | 0.85 (0.68–0.93) | 0.91 (0.81–0.96) | 0.85 (0.68–0.93) |
| | | SEM[d] | 2.89° | 2.95° | 2.33° | 2.74° |
| | | 2 SEMs[e] | 5.78° | 5.90° | 4.66° | 5.48° |
| | | MDC$_{95}$[f] | 8.01° | 8.18° | 6.46° | 7.59° |

**Notes.**
[a]ROM, Range of motion.
[b]ICC, Intraclass correlation coefficient.
[c]CI, Confidence interval.
[d]SEM, Standard error of measurement.
[e]2 SEMs, Two standard errors of measurement.
[f]MDC$_{95}$, Minimum detectable change at the 95% confidence level.

## RESULTS

Among the 49 subjects originally recruited, 19 were excluded based on the eligibility criteria: 16 had low back pain; 1 had a surgical instrument in the spine; and 2 had obvious scoliosis. Thus, 30 subjects were included in the current investigation.

The mean maximal flexion ROM measured by both raters ranged from 51°–52.30°. Also, the mean maximal extension ROM ranged from 16.50°–18.43° (Table 2). Measurement data from the intra-rater (within-day and between-day) and inter-rater (within-day) reliability analysis including the ICC with 95% confidence interval (CI), SEM, 2 SEMs, and MDC$_{95}$ are reported in Tables 3 and 4. Excellent intra-rater reliability was found with both the inclinometer and iPhone® for lumbar spine flexion and extension ROM ranging from 0.84–0.92 (Table 3). The results of the current study showed that the inter-rater reliability was good to excellent, ranging from 0.77–0.89 (Table 4). Concurrent validity for flexion and extension ROM between the gravity-based inclinometer and iPhone® app was excellent with $r$ values of 0.85 and 0.91, respectively (high direct relationship) (Fig. 7).

The 95% limits of agreement (Bland-Altman plots; Fig. 8) indicate that there are no differences between the values measured by the 2 instruments as the zero point lies within the range of differences between gravity-based inclinometer and iPhone® values. However, individual differences may range from the iPhone® being 6.26° greater (the upper limit of

**Table 4  Inter-rater reliability of gravity-based inclinometer and iPhone® app.**

| Inter-rater reliability | Gravity-based inclinometer | | iPhone® app | |
|---|---|---|---|---|
| | Flexion ROM[f] | Extension ROM[f] | Flexion ROM[f] | Extension ROM[f] |
| ICC[a] (95% CI[b]) | 0.77 (0.52–0.89) | 0.87 (0.73–0.94) | 0.85 (0.69–0.93) | 0.89 (0.76–0.94) |
| SEM[c] | 3.04° | 2.96° | 3.12° | 2.66° |
| 2 SEMs[d] | 6.08° | 5.92° | 6.24° | 5.32° |
| MDC$_{95}$[e] | 8.44° | 8.22° | 8.66° | 7.38° |

**Notes.**
[a] ICC, Intraclass correlation coefficient.
[b] CI, Confidence interval.
[c] SEM, Standard error of measurement.
[d] 2 SEMs, Two standard errors of measurement.
[e] MDC$_{95}$, Minimum detectable change at the 95% confidence level.
[f] ROM, Range of motion.

agreement) to 6.86° less (the lower limit of agreement) than the gravity-based inclinometer for lumbar flexion ROM. In lumbar extension ROM, the individual differences may range from the iPhone® being 4.87° greater (the upper limit of agreement) to 5.39° less (the lower limit of agreement) than the gravity-based inclinometer.

## DISCUSSION

The aim of the current study was to evaluate the intra-rater and inter-rater reliability, MDC$_{95}$, and concurrent validity of a gravity-based inclinometer (Vertex) and iPhone® app (TiltMeter© -advanced level and inclinometer) for measuring standing lumbar spine flexion and extension ROM in 30 asymptomatic subjects. The novelty of this investigation was that the within-day intra-rater reliability was evaluated as well. In addition, a new iPhone® app was utilized to measure isolated lumbar spine flexion and extension ROM.

Various disorders of the lumbar spine can affect the ROM, including ankylosing spondylitis, back strains, osteoarthritis, scoliosis, fractures, and spondylolisthesis (*Karnath, 2003*; *Danielsson, Romberg & Nachemson, 2006*; *McGregor, Cattermole & Hughes, 2001*). Therefore, measuring the ROM of the lumbar spine with valid and reliable instruments can help clinicians provide more accurate clinical assessment and intervention in patients with lumbar spine problems. A gravity-based inclinometer is an instrument used for measuring lumbar spine flexion and extension ROM (*Ng et al., 2001*; *Saur et al., 1996*). The reliability of the gravity-based inclinometer in the current study is consistent with previous studies, which have reported good to excellent ICC values when applying similar measurement procedures (*Kolber et al., 2013*; *Ng et al., 2001*; *Salamh & Kolber, 2013*; *Saur et al., 1996*; *Waddell et al., 1992*). Good to excellent reliability in this study is likely due to controlling the slippage of instrument on subjects' skin during movement via firm placement and also accurate spinal bony landmark detection. In addition, one rater was responsible for marking the subjects in all 3 sessions, this may minimize the confounding variables. Prior to testing, all the subjects were familiarized with the procedures through demonstration, and according to *Dankaerts et al. (2006)*, familiarity could enhance the reliability. The results also revealed that both raters had excellent intra-rater reliability. The raters were experienced orthopedic physiotherapists and this factor could be another reason of high

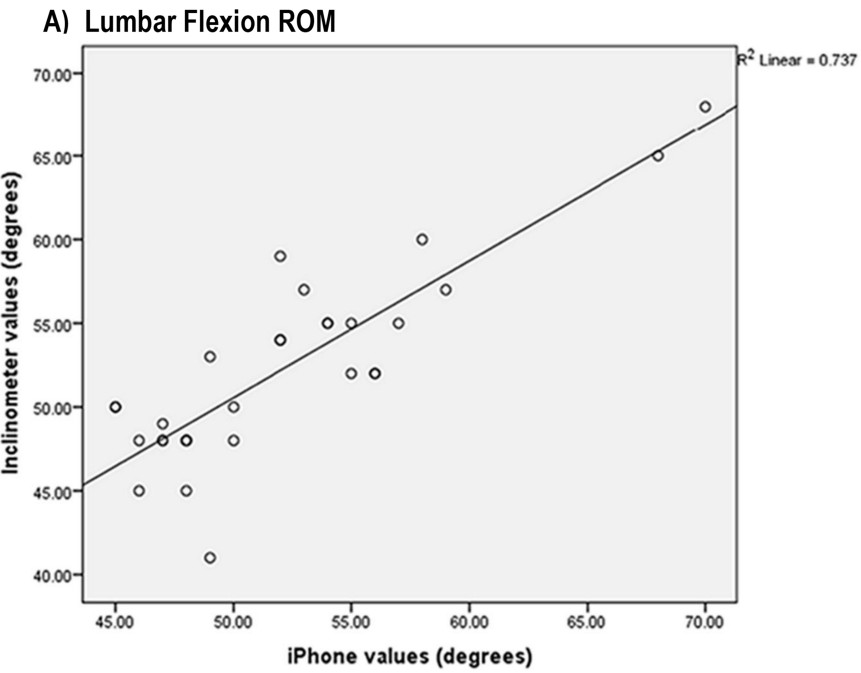

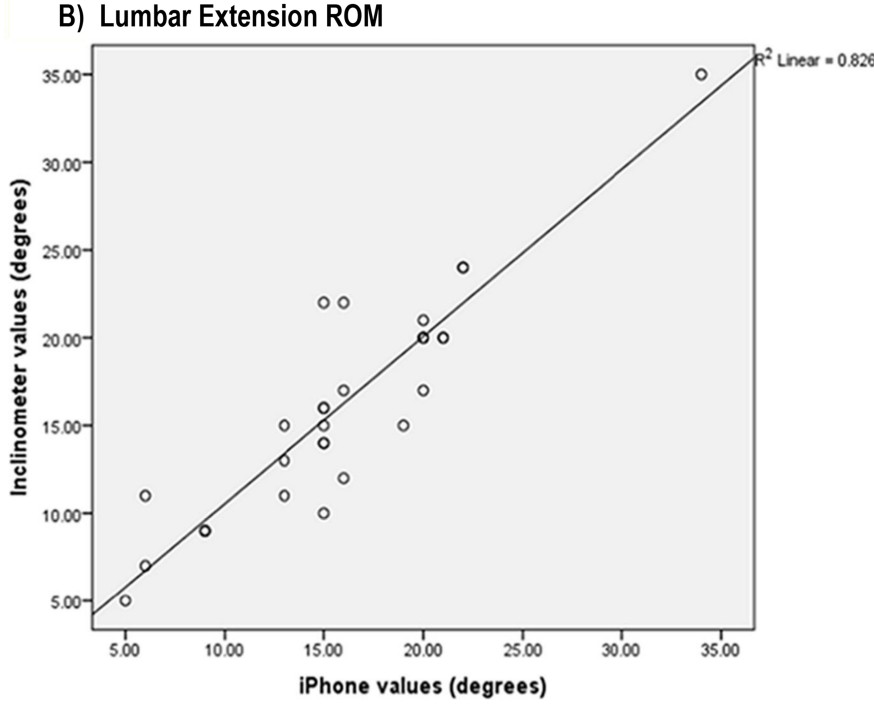

**Figure 7** Scatterplots showing relationships between the iPhone® and gravity-based inclinometer for maximum lumbar (A) flexion and (B) extension ROM.

A) Lumbar Flexion ROM

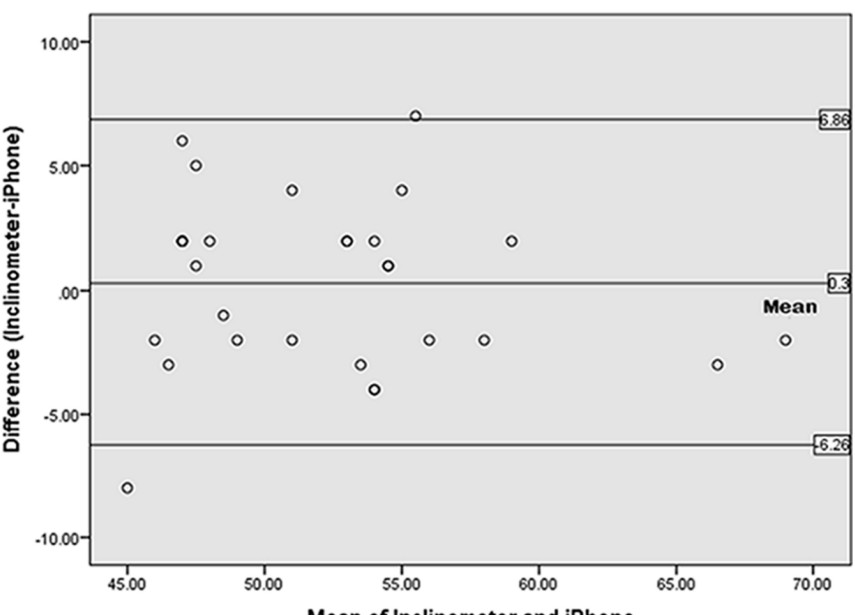

B) Lumbar Extension ROM

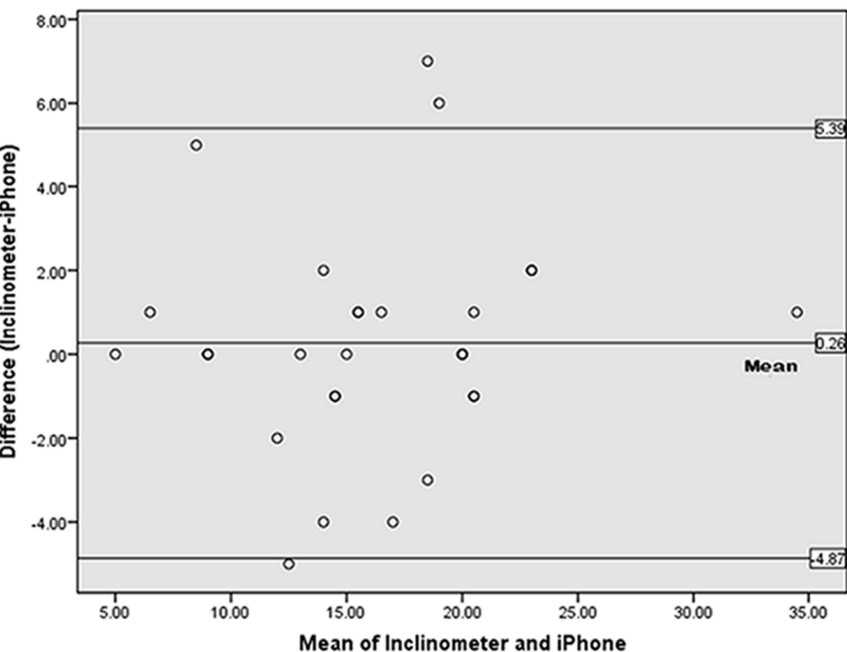

**Figure 8** Bland–Altman plots representing mean differences and 95% limits of agreement between inclinometer and iPhone® measurements of maximum lumbar (A) flexion and (B) extension ROM (degrees).

reliability. However, it should be noted that the reliability values cannot be extrapolated to less-experienced raters and further research is needed to investigate the reliability of the current iPhone® app in less-experienced raters.

There are few published studies measuring the reliability and validity of iPhone® apps (*Kolber et al., 2013*; *Salamh & Kolber, 2013*). In general, the results of this study are in agreement with *Kolber et al. (2013)* and *Salamh & Kolber (2013)*. Within-day intra-rater and inter-rater reliability of the iPhone® app were good to excellent. Unfortunately, to the best of the authors' knowledge, there are no published data with which to compare within-day intra-rater reliability findings. Similar to the gravity-based inclinometer, excellent reliability could also be due to firm placement, accurate spinal landmarks detection, minimizing the confounding variables, and familiarity. The mean descriptive measurement values of the 2 instruments were similar. Except in lumbar spine extension, the iPhone® app mean values were slightly lower than the gravity-based inclinometer mean values. The difference in the shape of the two measurement instruments (*Kolber et al., 2013*; *Salamh & Kolber, 2013*) and slight difficulty in maintaining constant skin contact of the iPhone® at the extremes of extension ROM may explain the difference between the mean values of the 2 instruments. However, the difference was not statistically significant to cause concerns to clinicians about the measurement of lumbar spine ROM using the iPhone® app (please see Bland-Altman plots). *Kolber et al. (2013)* measured isolated lumbar spine flexion ROM using an iPhone® model 4 (iHandy© level app) and a bubble inclinometer in 30 asymptomatic subjects. The mean flexion measured by iPhone® for 2 raters was 49.5°, and in our study, the mean flexion measured by both raters was 51.5°. However, there is a small difference (∼6.6°) between the mean flexion measured using inclinometers reported by *Kolber et al. (2013)* and our investigation. The difference in the shapes of inclinometers used in the 2 studies could be one reason for this difference. Also, we assessed each subject 3 times and the mean of 3 repetitions was analyzed, therefore, this can provide more reliable results. The comparison could not be made for lumbar spine extension ROM since *Kolber et al. (2013)* did not measure isolated lumbar spine extension ROM.

Validity refers to how well a measurement instrument actually measures the underlying outcome of interest (*Sullivan, 2011*). According to *Sullivan (2011)*, measurement instruments must be also valid for study results to be credible. Concurrent validity analysis of the current study showed high correlations between the inclinometer and the TiltMeter©-advanced level and inclinometer app with $r$ values $\geq 0.85$. The results of the validity are consistent with *Kolber et al. (2013)* and *Salamh & Kolber (2013)* studies, which have reported high $r$ values ($\geq 0.86$). However, the iPhone® app used in the previous studies was different from this study. The findings of the investigation confirmed the hypotheses that the TiltMeter© -advanced level and inclinometer app is a valid and reliable instrument for measuring standing lumbar spine flexion and extension ROM. Another point worth mentioning is that the external validity of the results could be limited since the rater A (MRP) was only responsible for marking the subjects.

One advantage of using the iPhone® app over the gravity-based inclinometer is that the app is available anytime, anywhere and can be downloaded for free from Apple's App Store. Moreover, this app is also available for free for Android smartphones on

AppCrawlr (http://appcrawlr.com/). However, it is important to consider limitations related to smartphone use. Clinicians may be reluctant to use their personal smartphone for assessment, because there would be a direct contact between the patient's skin and the smartphone (*Kolber et al., 2013*; *Salamh & Kolber, 2013*). In addition, sometimes smartphones hang or freeze unexpectedly and would therefore, obstruct or interfere with the assessment.

As the number of smartphone users has been increasing since the last decade, utilizing a valid and reliable app instead of traditional instruments can provide an easy, simple and cost-effective measurement of joint ROM.

### Study limitations and future research recommendations

The subjects of the current study were asymptomatic with approximately normal body mass index (BMI). Thereby, the generalizability of these findings is limited and the data obtained from the asymptomatic subjects is not representative of the population with low back pain. Further research is needed to evaluate symptomatic subjects or individuals with different body morphology, because excess adipose tissue may affect the ability to accurately detect bony landmarks for instrument placement. In addition, only flexion–extension ROM was evaluated and not lateral flexion or axial rotation. Because in each of the 3 sessions the subjects were measured 3 times by the 2 raters at the $T_{12}$–$L_1$ and $S_1$–$S_2$ spinal levels using the 2 instruments, hence, we thought that the measurement of movements in other planes could cause microtrauma to the subjects. As mentioned in the discussion, for controlling confounding variables and improving consistency among the placement of the 2 instruments, one rater (MRP) was responsible for marking the subjects; therefore, the external validity could be limited.

Last but not least, future research is encouraged to assess the reliability and validity of this app for measuring the ROM of other joints as well. Such an analysis would give insight on the application of smartphone devices in physiotherapy and orthopedic assessment.

## CONCLUSIONS

Measurement of joint ROM is a part of physiotherapy and orthopedic assessment of various pathologies. Hence, numerous instruments have been introduced for this purpose (e.g., goniometers, inclinometers, measuring tapes, etc.). Smartphone inclinometer apps have been developed in recent years which allow clinicians to evaluate joint ROM more quickly and easily. The TiltMeter$^{©}$ -advanced level and inclinometer app is an inclinometer app of iPhone$^{®}$. This investigation demonstrated that the app possesses good to excellent reliability (ICC $\geq$ 0.77) and concurrent validity with a gravity-based inclinometer ($r \geq 0.86$) for measuring standing isolated lumbar flexion and extension ROM. According to the findings of the study, it seems that the app can be used for the measurement of lumbar spine flexion and extension ROM.

## ACKNOWLEDGEMENTS

The authors would like to thank all the subjects who participated without whose support this study would not have been possible.

### Funding
The authors received no funding for this work.

### Competing Interests
The authors declare there are no competing interests.

### Author Contributions
- Mohammad Reza Pourahmadi conceived and designed the experiments, performed the experiments, analyzed the data, contributed reagents/materials/analysis tools, wrote the paper, prepared figures and/or tables, reviewed drafts of the paper.
- Morteza Taghipour conceived and designed the experiments, performed the experiments, analyzed the data, reviewed drafts of the paper.
- Elham Jannati performed the experiments, prepared figures and/or tables, reviewed drafts of the paper.
- Mohammad Ali Mohseni-Bandpei conceived and designed the experiments, analyzed the data, reviewed drafts of the paper.
- Ismail Ebrahimi Takamjani and Fatemeh Rajabzadeh analyzed the data, reviewed drafts of the paper.

### Human Ethics
The following information was supplied relating to ethical approvals (i.e., approving body and any reference numbers):

Approval for the study was obtained from the Ethics Committee at the University of Social Welfare and Rehabilitation Sciences (801-2-6189).

### Data Deposition
The raw data has been supplied as Data S1.

### Supplemental Information
Supplemental information for this article can be found online at http://dx.doi.org/10.7717/peerj.2355#supplemental-information.

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
