# Peer review of "Reliability and validity of an iPhone® application for the measurement of lumbar spine flexion and extension range of motion"

_PeerJ, doi:10.7717/peerj.2355_

## Round 0.1 · original submission · Major Revisions

· Academic Editor

Major Revisions

All three reviewers have described some of the positive aspects of the study and paper, while also highlighting some issues that require attention. Please provide a detailed response to each of these reviewers comments and amend the manuscript accordingly, so that the manuscript can be given greater consideration for publication.

·

Basic reporting

Article: Reliability and validity of an iPhone® application for the measurement of lumbar spine flexion and extension range of motion

The purpose of this study was to investigate the reliability and validity of an iPhone® app (TiltMeter© -advanced level and inclinometer) for measuring standing lumbar spine flexion–extension ROM in asymptomatic subjects.

The article is very well written and clearly presented. The article clearly contributes new knowledge and provides a significant contribution to our understanding of the application of the iPhone® app for measuring standing lumbar spine flexion–extension ROM in asymptomatic subjects.


Minor amendments needed:

Line 78: Chaffin reference – list all 3 authors (see PeerJ reference guidelines)
Line 80: Nielsen reference – list all 3 authors
Line 102: Mellin reference – list all 3 authors
Line 154 and 263: Change ‘table’ to ‘Table’ (check PeerJ guidelines)
Line 219: Change ‘form’ to ‘from’.
Line 323: The authors may consider removing Table 2 as generally tables and figures are not referred to in the discussion.
Table 2: Consistency needed with decimal places throughout the table.

Experimental design

The authors provide a detailed and robust experimental design with sufficient information to be reproducible by another investigator. The research has been conducted in conformity with the prevailing ethical standards in the field.

Validity of the findings

The presented data are robust and statistically sound. The data on which the conclusions are based are provided and supported by the results.

·

Basic reporting

The authors have done a great job overall, but I feel as if some of the English can be improved throughout the manuscript, as to improve flow and clarity:
- Throughout the manuscript, I would prefer that the authors consistently utilize Oxford commas in order to help prevent ambiguity.
- Line 29: Please change “is always considered as an” to “is often considered to be”
- Line 32: I think you mean ‘accelerometer’, not ‘acceleration’. If so, please correct this.
- Line 40-41: I think this would read more clearly if it were reworded slightly; for example: A convenience sample of 30 asymptomatic adults (15 males; 15 females; age range = 18–55 years) were recruited for this study.
- Line 49: Please change “were the same as” to “were carried out in the same manner as”.
- Line 50: Please change “All the measurements” to “All of the measurements”; alternatively, you may remove “the”.
- Line 54: Please add “both” before “the gravity-based inclinometer”
- Line 56: Please add “from” after “ranged”
- Line 57: Please add “from” after “and”
- Line 71: Please change to either “whether an abnormal limitation exists”
- Line 75: Please change to “The method or instrument…”
- Line 96: Please remove the “the” preceding “intraclass correlation coefficients”
- Line 99: Please change “reporting” to “reported”
- Lines 109–110: Please pluralize accelerometer, magnetometer, gyroscope, and position
- Line 112: Please change “and” to “which”
- Line 124: Please change “result” to “results” and “indicated” to “Indicate”
- Lines 125–126: Please place the ICC values in parentheses, as to separate them from the sentence for clearer reading.
- Line 128: Please change “between-days” to “between-day”
- Line 147: Please add the word “the” before “of”
- Line 175: Please reword to “6 years of experience”
- Line 300: Please change “because of” to “due to the”
- Line 308: Please remove “studies”
- Line 316: Please “and little difficulty to maintain the constant” to “and slight difficulty in maintaining constant”
- Line 317: Please change “cause a” to “explain the”
- Line 319: Please change “In Kobler et al. (2013) study,” to “Kobler et al. (2013) found” or something similar.
- Line 321: Please remove “of”
- Line 324: Please reword to “using inclinometers reported by Kobler et al. (2013) and our investigation”
- Line 347: Please insert “an” before “easy”
- Line 361: Please reword to “the ROM of other joints as well”

There are some minor issues with some phrasing and referencing in the manuscript:
- Lines 291–293: The referenced papers do not suggest a causal, prospective, or contributory relationship between range of motion and LBP. The Zafereo paper was correlational/cross-sectional and does not examine lumbar ROM. The Chaitow reference was a series of commentaries, one of which argues against the postural-structural-biomechanical model of LBP, and within, references studies that suggest that there is no relationship between spinal ROM and pain. I would like to see this this sentence reworded (i.e., also acknowledging biopsyschosocial contributions) or updated with more relevant references.
- Please review the citations and references to ensure all spelling is correct. For example, throughout the manuscript, ‘Kolber’ and ‘Kobler’ are used interchangeably.

The manuscript is well-structured and is in line with PeerJ’s guidelines. Although arguably atypical (for PeerJ), I think the highlights/takeaway section is great.

Figures and tables
- Figures 7 and 8: Is there any way these can be made clearer, or perhaps recreated in a different program? They are somewhat pixelated and blurry.
- Figure 2: A screenshot of the app would be clearer and more appropriate than a picture of the iPhone.
- Tables: Please submit each table individually
- Table 1: Please change “Weight” to “Body Mass”

Experimental design

Because nothing was mentioned about participants potentially having a history of low back pain, I think it would be beneficial to justify their inclusion. This reference may be of interest: Esola, M. A., McClure, P. W., Fitzgerald, G. K., & Siegler, S. (1996). Analysis of lumbar spine and hip motion during forward bending in subjects with and without a history of low back pain. Spine, 21(1), 71-78.

Line 176: Can you please reword the sentence pertaining to the warm-up and expand upon it? How were the participants instructed? How was 45º measured, or was it “eyeballed”?

Line 216–217: In the clinic (and in most studies), marking an individual that is going to be tested will likely be completed by the same person that will be completing the measuring. By having a different individual doing the marking, yes, you are isolating your outcome measures, but at the same time, you are decreasing external/ecological validity. It can be argued that this is inappropriate for interrater reliability. Although Kolber’s papers did the same, I would like to see this addressed in the discussion.

Why was only one rater (rater A, MRP) used to for the calculation of test-retest (intrarater) reliability? Because you have the data for both raters, I’d like to see a combined intra-rater reliability, as this would make the values more robust (see Beardsley et al. (2016)’s paper on pelvic tilt as an example). On a similar note, I think it should be noted in your discussion that the reliability values cannot be extrapolated to less-experienced or less-trained raters.

Validity of the findings

Lines 276–278: What are the “mean differences” in this case? 0.26 and 0.3 are close to zero, but are not zero. Please elaborate.

Lines 301–303: As mentioned above, having one rater perform the marking is both a strength and a weakness. Please address this.

Lines 318–319: This sentence needs to be reworded and also brings up some questions. How were these values compared? The statistical procedures do not include the use of a t-test or similar procedure for testing differences between groups, nor was this reported in the results or any of the tables/figures. Please include this in the statistical procedures and results.

Reviewer 3 ·

Basic reporting

No comments

Experimental design

See comments to authors

Validity of the findings

See comments to authors

Comments for the author

Reliability and validity of an iPhone® application for the measurement of lumbar spine flexion and extension range of motion
Thank you for the opportunity to evaluate this paper. The authors have evaluated the reliability of a smartphone App to measure standing lumbar spine flexion and extension in asymptomatic subjects and compared this method to an inclinometer method. Generally the paper is well written although the introduction could be written more succinctly and provide a better synthesis of previous research findings. Discussion was well written and answered the questions posed without overstating the findings.

Specific comments
Line 29: omit ‘as’
Line 32: Word missing? i.e. ‘…goniometers, and etc’.
Line 34: Purpose says to ‘assess reliability and validity’ – but next sentence (under Design heading) only mentions reliability. Also suggest rewording the sentence in Design section.
Line 40: Check journal’s requirements – ‘participants’ is generally preferred to ‘subjects’.
Line 43: my suggestion would be to make the methods section in the abstract more succinct. Include how MDC was calculated.
Line 70: Avoid starting sentences with “This” – make it clear to what “this” refers
Line 72: Suggest: “Measurement of lumbar spine ROM”
Line 79 and Line 114: I would like to see these two paragraphs written more as a synthesis of the literature rather than a list of results of previous studies.
Line 105: Sentence needs rewording.
Line 129: I am more familiar with using MDC, not MCD, for minimal detectable change?
Line 154: I will be guided by the Editor but I am not sure the baseline characteristics needs to be in a Table? If so, I would put the variables in the first column – one line tables can look a bit odd!
Line 172: I appreciate the authors acknowledging the limitation of crook lying rotations as a ‘warm-up’ exercise but I am wondering why they did not choose a flex/ext based warm up given it was flex/ext they were intending on measuring?
Line 184 and Line 200: ‘Shoulder width’ not ‘shoulder length’
Line 185: ‘Intercristal’?
Line 252: Avoid one sentence paragraphs
Discussion was generally well written – not convinced about Line 358 is all!
Line 377: Good to excellent.

---

## Round 0.2 · Minor Revisions

· Academic Editor

Minor Revisions

The reviewers were very happy with the amendments you've made to the initial submission, but just request a few very minor amendments before this can be accepted for publication.

·

Basic reporting

See general comments

Experimental design

See general comments

Validity of the findings

See general comments

Comments for the author

The authors have done well addressing the comments of the reviewers.

Just some final minor amendments are needed (see below):

Abstract:
Recent smartphones have been equipped with accelerometer and magnetometer, which, through specific software applications (apps) can be used for inclinometric functions.

Change to: accelerometers and magnetometers

Subjects. A convenience sample of 30 asymptomatic adults (15 males; 15 females; age range = 18–55 years) were recruited August 2015 and December 2015.

Add ‘between August 2015 and December 2015’.

Discussion
Please end this sentence appropriately:
Various pathologies can affect the ROM of the lumbar spine, such as ankylosing spondylitis, back strains, and etc (Karnath, 2003).

Expand on last (standalone) sentence:
Last but not least, future research is encouraged to assess the reliability and validity of this app for measuring the ROM of other joints as well.

Example: Such an analysis would give insight……

References:
Remove comma from in-text references (when out of brackets) e.g. Saur et al. (1996) not Saur et al., (1996)

Remove abbreviated ‘&’ from in-text references (when out of brackets) e.g. Salamh and Kolber (2013); not Salamh & Kolber (2013)

Reference list: All Journals should be in capitals e.g. ‘Journal of Spinal Disorders & Techniques’; not ‘Journal of spinal disorders & techniques’.

·

Basic reporting

I would like to thank the authors for adequately addressing all of my comments and congratulate them on a job well done.

At this point, I think only two small changes need to be made:
1) Please update the reliability values in the abstract to reflect the ones in the manuscript and Table 3 (i.e., with Rater B's values).
2) In Table 1, please get rid of the "± SD" in parentheses. Although nitpicky, I think it's important to distinguish between units and what you are reporting. It is mean ± SD that you are reporting, while the units for both the mean and SD are in years, kg, cm, or kg/m^2. I think it would be best to have the table head read: Age (years), Body mass (kg), etc., and have a footnote stating that values are presented as mean ± SD.

Experimental design

No comments

Validity of the findings

No comments

---

## Round 0.3 · accepted · Accept

· Academic Editor

Accept

On behalf of the reviewers, I'd like to thank you for making the necessary adjustments to this manuscript which we are now happy to accept for publication.

·

Basic reporting

See below

Experimental design

See below

Validity of the findings

See below

Comments for the author

The authors have done well and have addressed the issues that were raised.

·

Basic reporting

I have no further comments.

Experimental design

I have no further comments.

Validity of the findings

I have no further comments.